# Tetrahydrocannabinol Reduces Hapten-Driven Mast Cell Accumulation and Persistent Tactile Sensitivity in Mouse Model of Allergen-Provoked Localized Vulvodynia

**DOI:** 10.3390/ijms20092163

**Published:** 2019-05-01

**Authors:** Beebie Boo, Rohit Kamath, Erica Arriaga-Gomez, Jasmine Landry, Elizabeth Emanuel, Sookyong Joo, Marietta Saldías Montivero, Tijana Martinov, Brian T. Fife, Devavani Chatterjea

**Affiliations:** 1Biology Department, Macalester College, Saint Paul, MN 55105, USA; beebieboo@gmail.com (B.B.); rkamath1611@gmail.com (R.K.); earriag1@macalester.edu (E.A.-G.); jasminealandry@gmail.com (J.L.); eemanuel@macalester.edu (E.E.); sookyong.joo@gmail.com (S.J.); msaldias@macalester.edu (M.S.M.); 2Center for Immunology, University of Minnesota, Minnesota, MN 55455, USA; tijana.martinov09@gmail.com (T.M.); bfife@umn.edu (B.T.F.)

**Keywords:** mast cells, dinitrofluorobenzene, hypersensitivity, chronic pain, vulvodynia, Δ^9^-tetrahydrocannabinol

## Abstract

Vulvodynia is a remarkably prevalent chronic pain condition of unknown etiology. An increase in numbers of vulvar mast cells often accompanies a clinical diagnosis of vulvodynia and a history of allergies amplifies the risk of developing this condition. We previously showed that repeated exposures to oxazolone dissolved in ethanol on the labiar skin of mice led to persistent genital sensitivity to pressure and a sustained increase in labiar mast cells. Here we sensitized female mice to the hapten dinitrofluorobenzene (DNFB) dissolved in saline on their flanks, and subsequently challenged them with the same hapten or saline vehicle alone for ten consecutive days either on labiar skin or in the vaginal canal. We evaluated tactile ano-genital sensitivity, and tissue inflammation at serial timepoints. DNFB-challenged mice developed significant, persistent tactile sensitivity. Allergic sites showed mast cell accumulation, infiltration of resident memory CD8^+^CD103^+^ T cells, early, localized increases in eosinophils and neutrophils, and sustained elevation of serum Immunoglobulin E (IgE). Therapeutic intra-vaginal administration of Δ^9^-tetrahydrocannabinol (THC) reduced mast cell accumulation and tactile sensitivity. Mast cell-targeted therapeutic strategies may therefore provide new ways to manage and treat vulvar pain potentially instigated by repeated allergenic exposures.

## 1. Introduction

Vulvodynia is a complex, chronic vulvar pain condition without a clear underlying etiology that affects between 10 to 28% of women of reproductive age as assessed primarily in US urban populations [1,2]. Women who experience this particular form of genital pain often struggle to receive a diagnosis and are frequently unable to find an effective treatment. Along with painful sensitivity to touch, a clinical diagnosis of vulvodynia often includes an increase in mast cells, local overgrowth of nerves [3], and altered inflammatory responses of vulvar tissue-derived fibroblasts to in vitro yeast antigen stimulation [4,5]. A history of seasonal and cutaneous allergies doubles the risk for vulvodynia [6], and its incidence is linked with exposure to household and workplace cleaning chemicals [7]. We have previously reported that repeated exposures to the hapten oxazolone on the labiar skin of ND4 (Notre Dame re-derived) Swiss Webster female mice recapitulate key clinical symptoms, i.e., painful ano-genital sensitivity to touch, increased nerve growth and an accumulation of mast cells [8,9]. We also found that local mast cell depletion by intra-labiar injections of basic secretagogue compound 48/80 (c48/80) decreased pain sensitivity [8]. Here, we investigated whether this effect was hapten-specific by exposing ND4 Swiss female mice to dinitrofluorobenzene (DNFB) instead of oxazolone. Allergic exposure to DNFB in mice induces contact hypersensitivity [10] and airway hyperresponsiveness [11], both of which are mast cell driven. Dermal mast cells and dendritic cells are the first responders to DNFB in the skin and orchestrate the subsequent T cell driven inflammation [12]. Given the importance of mast cells in allergen-driven vulvar pain, the capacity for specific induction of mast cell driven responses was critical to our choice of DNFB as the hapten allergen for these experiments. Furthermore, we used a non-traditional saline vehicle so that we could apply DNFB challenges inside the vaginal canal of ND4 mice. Mouse models of contact hypersensitivity typically use harsher, drying organic solvents such as ethanol or acetone to deliver hapten allergens onto the skin [13,14]. Given that clinical tests for vulvodynia [1] and associated inflammation [4,5] are conducted on the mucosal-like vulvar tissue, we adapted our protocol to apply DNFB dissolved in normal saline within the vaginal canal of mice that had been previously sensitized to DNFB on the shaved flank and measured changes in ano-genital sensitivity to touch and pressure, as well as inflammatory responses in the affected tissue. Finally, we assessed the effect of therapeutic intra-vaginal applications of Δ^9^-tetrahydrocannabinol (THC) on the abundance of mast cells and painful sensitivity.

## 2. Results

### 2.1. Ten Exposures to Dinitrofluorobenzene (DNFB) Dissolved in Saline on the Labiar Skin of Previously Sensitized ND4 Swiss Mice Induce Local and Systemic Inflammatory Responses

To demonstrate that allergic tissue changes occur in response to DNFB dissolved in saline instead of a more typical, harsher vehicle such as acetone, we measured mast cell and T cell infiltration as well as changes in inflammatory cytokines in the allergic skin of previously sensitized ND4 female mice after 10 DNFB challenges. Labiar mast cell density increased in flank-sensitized mice challenged with 10 daily topical DNFB administrations to the labiar skin compared to mice challenged with saline; there was a four-fold increase one day after the completion of challenges, and a seven-fold increase at 21 days (Figure 1A,B). Interestingly, serum IgE levels were significantly elevated compared to controls challenged only with saline 1 day after 10 DNFB challenges and remained so 42 days after challenges were completed (Figure 1C). Tissue eosinophil peroxidase and tissue myeloperoxidase levels increased sharply in DNFB-challenged mice vs. saline-treated controls one day after 10 challenges and resolved to levels indistinguishable from controls by 21 days after challenge cessation (Figure 1D,E). One day after the 10th DNFB administration, flow cytometric analysis of dissociated flank skin revealed a significant accumulation of CD3^+^, CD4^+^, activated CD4^+^CD44^+^, CD8^+^, and resident memory CD8^+^CD103^+^ in hapten-challenged but not in control mice (Figure 1F). mRNA transcripts encoding *IL-6* and *IFN-γ* were upregulated 8 and 3-fold respectively in DNFB- vs. saline-challenged mice 1 day post challenge cessation. *IL-6* mRNAs remained slightly upregulated over controls at 21 days after 10 DNFB challenges (Figure 1G). *IFN-γ*, which has been shown to drive mast cell-mediated immune responses [15], was increased to 15-fold in DNFB- vs. saline-treated mice 21 days after 10 labiar challenges. All visually detectable signs of irritation (slight redness) resolved within 2 days of cessation of DNFB challenges. Since the inflammatory effects of DNFB in more typical organic solvent vehicles have, historically, been characterized mostly in inbred mouse strains, we verified that outbred ND4 female mice sensitized on the flank and subsequently challenged on the labiar skin with DNFB dissolved in acetone showed increases in local mast cell abundance and abundances of transcripts encoding *IL-6* and *IFN-γ* (Appendix AA–C). However, we noticed that the harsher acetone solvent caused more dryness of the labiar skin compared to the saline vehicle we used for these studies.

### 2.2. Ten DNFB Administrations on the Labiar Skin, or in the Vaginal Canal, of Previously Sensitized ND4 Swiss Mice Provoke Persistent Vulvar Mechanical Tactile Sensitivity

Since repeated exposures to DNFB dissolved in saline provoked inflammatory responses in keratinized labiar skin, we next assessed the potential for DNFB/saline to provoke tactile ano-genital sensitivity when administered on the labiar skin or, separately, into the vaginal canal of ND4 mice. We measured tactile sensitivity using an electronic von Frey meter at serial timepoints after 10 DNFB challenges applied on the labiar skin or pipetted into the vaginal canal in previously sensitized ND4 Swiss mice. One day after cessation of challenges, mice repeatedly exposed to DNFB topically on the labiar tissue (Figure 2A) or directly into the vaginal canal (Figure 2B) showed significantly increased tactile sensitivity compared to mice similarly exposed to 0.9% saline. DNFB treated mice had a ~60% decrease from baseline withdrawal measurements after DNFB exposure in both tissues one day after the 10th challenge. Mice treated with saline had a ~30% decrease in withdrawal threshold to touch. This localized hapten-dependent increase in tactile sensitivity persisted past 21 days post 10th DNFB exposure on the labiar tissue and in the vaginal canal long after resolution of overt inflammation in the tissues. Both DNFB and saline challenged mice returned to baseline withdrawal values by 42 days after the last DNFB exposure. At all timepoints after the 10th challenge, mice treated with saline did not show painful responses; they had a percent decrease from baseline value lower than the hyperalgesia cutoff of a 33% difference between average baseline and post-treatment values described by us and others [8,9,16]. ND4 female mice sensitized on the flank and subsequently challenged on the labiar skin with DNFB dissolved in acetone also displayed heightened genital sensitivity to pressure one day after the cessation of challenges (Appendix AD).

### 2.3. Ten DNFB Exposures into the Vaginal Canals of Previously Sensitized ND4 Swiss Mice Induce Local and Systemic Inflammatory Responses

In addition to provoking tactile ano-genital sensitivity when applied inside the vaginal canal, repeated administration of DNFB into the vaginal canal of ND4 mice produced inflammatory changes in abundances of cells and molecules similar to those we described above in labiar skin. Mast cells accumulated at ~3-fold higher levels in the vaginal canal tissues of DNFB-challenged vs. saline-challenged mice and persisted through 21 days after 10 hapten administrations (Figure 3A,B). mRNA transcripts encoding *IL-6* were slightly upregulated at 21 days after 10 DNFB challenges while *IFN-γ* transcripts were upregulated ~4-fold one day after the 10th challenge and ~2-fold at 21 days (Figure 3C). Serum IgE, tissue eosinophil peroxidase and tissue myeloperoxidase levels were all significantly elevated in DNFB- vs. saline-challenged mice a day after 10 challenges to the vaginal canal (Figure 3D–F). Mast cells (Figure 3G), B cells, CD4^+^ T cells, and CD8^+^ T cells (Figure 3H) were elevated in the draining iliac lymph nodes at 1 day, 3 days, and 7 days in mice challenged 10 times daily with DNFB in the vaginal canal as compared to saline-challenged controls.

### 2.4. Repeated Δ^9^-Tetrahydrocannabinol (THC) Application Reduces Tactile Sensitivity and Mast Cell Density after 10 Exposures to DNFB in the Vaginal Canals of Previously Sensitized ND4 Mice

Gaffal and colleagues have demonstrated the anti-inflammatory activity of THC, the main psychoactive phyto-cannabinoid of cannabis, in DNFB-driven dermatitis in mice [16]. Others have elucidated the effects of THC on mast cell function in oxazolone-driven dermatitis [17]. Here we tested the potential of THC, administered topically inside the vaginal canals of DNFB-challenged mice, to alleviate tactile sensitivity and reduce mast cell accumulation. We pipetted the cannabinoid THC (in saline) directly into the vaginal canal of sensitized ND4 Swiss mice for six consecutive days starting one day after 10 DNFB administrations within the vaginal canal. One day after six THC applications i.e., on day 7 after 10 DNFB challenges, mice showed decreased sensitivity compared to mice that were challenged with DNFB but not treated with vaginal THC. (Figure 4A). THC treated mice had a ~30% decrease from baseline withdrawal values while the mice with no THC treatment showed a 60% decrease from baseline withdrawal thresholds. The decrease in sensitivity in the THC treated mice was accompanied by a decrease in local mast cell density (Figure 4B,C).

## 3. Discussion

The contributions of mast cells to pain pathologies have been well studied in recent years [18]. These “tunable” effectors of tissue pathology [19] are now known to contribute to protective painful responses to noxious stimuli [20,21] as well as to acute and chronic pain conditions such as migraines [22], sickle cell pain [23], irritable bowel disorders [24] and pain associated with neuropathic and neurological diseases [25,26]. Mast cell increases in vulvar biopsy samples from vulvodynia patients were first reported over a decade ago [3,27] but are not ubiquitous in all patients [28]. Despite its surprisingly high prevalence vulvodynia remains a diagnosis of exclusion [29] and its pathophysiology is poorly understood. Medical histories of yeast infections [30], other urinogenital infections [31], depression and anxiety [32], and allergies [6] have all been associated with elevated vulvodynia risk. However, these possibly overlapping but likely importantly distinct etiologies and relevant underlying mechanisms have not been differentiated and there are no known biomarkers to appropriately diagnose sub-types of vulvar pain. As a result, efficacious, targeted therapies are yet to be developed for the management of this condition.

There are very few established pre-clinical models of persistent genital pain that can be used to investigate disease process, biomarkers, and therapies. We have, both previously [8,9] and here, adapted laboratory hapten-based mouse contact hypersensitivity protocols to show that mast cell accumulation following repeated allergen exposures lasts weeks beyond the resolution of overt inflammation. It has been previously shown that mast cell accumulation in the airways following exposure to the allergen ovalbumin is dependent on the presence of regulatory T cells [33] and mast cell-mediated tissue remodeling in chronic asthma-like disease in mice is dependent on IFN-γ [34]. Here we’ve reported the early accumulation of regulatory CD4^+^CD25^+^ T cells in DNFB/saline-challenged skin that may potentially serve to recruit mast cells to the site of the reaction, and also infiltration of resident memory CD8^+^CD103^+^ cells that can potentially serve as a key source of IFN-γ [35]. Importantly, IFN-γ and IL-6 are elevated both in the local tissue of challenged mice in our experiments, and in biopsies of vulvodynia patients versus controls [5]. The systemic increase in IgE we measured following DNFB challenges is also notable since IgE can serve as a survival factor for mast cells [36] along with IL-6 [37]. Interestingly while acute inflammatory markers such as neutrophil and eosinophil infiltration resolved rather rapidly after the cessation of DNFB challenges, mast cell accumulation, increase in IFN-γ mRNAs and heightened serum IgE levels persisted three weeks or longer along with detectable painful responses showing strong association between mast cell accumulation and genital pain (Figure 5).

Administration of DNFB using a saline vehicle also allowed us to apply allergen directly into the vaginal canal of ND4 mice to simulate contact dermatitis-driven exposures and inflammation in mucosal tissues that are more similar to the vestibular tissues associated with a diagnosis of vulvodynia [1]. We saw similar persistent tactile pain responses and mast cell accumulation, as well as increases in IFN-γ mRNAs following intra-vaginal applications of DNFB/saline in ND4 mice. Wang et al. observed increases in mast cells in the draining lymph node after dermal DNFB/acetone exposure [38]; these mast cells were critical to subsequent T cell recruitment and activity. After intra-vaginal DNFB/saline administration, mast cells expanded in the iliac lymph nodes of ND4 mice between 1 and 7 days. This, along with increases in serum IgE and early spikes in eosinophil and neutrophil activity in the mucosal tissue of the vaginal canal confirms that intra-vaginal administration of DNFB in a novel saline vehicle induced allergic inflammation that was accompanied by persistent painful sensitivity to touch.

We also saw a modest increase in the density of cutaneous sensory nerves in the vaginal canal and labiar skin tissues of sensitized mice challenged with DNFB vs. control mice challenged with saline (Appendix A). Nerve Growth Factor (*NGF*) and Cell Adhesion Molecule -1 (*CADM1*) mRNAs are slightly upregulated in labiar skin of sensitized mice after three challenges with DNFB dissolved in acetone (Appendix AE). Mast cells and nerves often jointly regulate tissue inflammation and pathology [18,39]; mast cells can secrete *NGF* [40] and *CADM1* is an important component of mast cell-nerve synapses that regulate the pathophysiology of atopic dermatitis lesions [41]. Therefore, it is possible that the accumulation of mast cells in DNFB/saline-treated tissues of the vaginal canal modulates increased density and activation of local nerves contributing to the painful response to touch and pressure.

While we see tissue mast cell numbers increase following repeated applications of both oxazolone [8,9] and DNFB (using either acetone or saline as vehicles), Farmer and colleagues [16] report that in their model of genital pain provoked by repeated vulvovaginal candidiasis, no mast cell increases were detected in the vaginal canal tissues. Therefore, non-normative accumulation of mast cells in vulvodynia patients may be a biomarker for an allergy-driven etiology of vulvar pain but not vulvar pain that stems from other root causes such as repeated yeast infections. Administration of mast cell granule stabilizer cromolyn sodium has been shown to have no significant therapeutic benefit for patients with vulvar vestibulitis who show increased mast cell numbers by biopsy [42]. This is concordant with our observation that while mast cells increase in numbers at the painful site, their contribution to the pain pathology appears to be driven by their presence rather than degranulation. We have, in fact, previously shown that local, transient removal of accumulated mast cells via secretagogue c48/80-mediated degranulation is associated with a transient *relief* from painful responses to pressure [8]. Here, we show that daily therapeutic topical administration of THC in the vaginal canal for six days resulted in a decrease in painful sensitivity to pressure in the ano-genital region as well as reduced local mast cell density in the vaginal canal tissue of previously sensitized mice challenged with 10 doses of DNFB in the vaginal canal. While the actions of endocannabinoids and cannabis-based medicines on neurological pain pathways are well established [43], here we observed changes in mast cell numbers following THC administration. Cannabinoid receptor-1 (CB1R) agonists have been shown to directly downregulate activity of cultured RBL-2H3 mast cells and mast cell activity in oxazolone-dermatitis [18] and topical THC has been shown to reduce symptoms of DNFB-dermatitis independent of CB1 and CB2 receptors [16]. Cannabinoid receptor-specific mechanisms have also been shown to mitigate mast cell activation, neurogenic inflammation and pain associated with sickle cell disease [44]. Singh and colleagues showed that a CB2 receptor agonist ameliorated dextran sodium sulfate-driven colitis in IL10−/− mice along with a reduction of mast cell numbers in the lamina propria and mesenteric lymph nodes although the specific mechanisms of cannabinoid-driven mast cell reduction have yet to be elucidated [45]. With recent changes in regulatory practices around therapeutic and recreational marijuana use, the use of cannabis products for medical purposes including pain relief is rapidly expanding [46]. A recent meta-analysis of randomized clinical trial reports [47] concluded that cannabis-based medicines might be effective for chronic pain but are more efficacious for neuropathic pain and that adverse effects are more likely with oral and mucosal administration rather than inhalation. Given these and other caveats, more thorough risk-benefit analyses must be conducted before cannabis-based therapies can be recommended more broadly. Our results here suggest, however, that local, topical cannabinoid administration could potentially reduce pathological mast cell accumulation that drives chronic vulvar pain, and therefore deserves more thorough examination in pre-clinical and clinical settings.

Vulvodynia is a complex, multi-factorial pain condition that likely stems from a wide variety of underlying causes and unfolds in the diverse inflammatory contexts of individual experiences and medical histories. With the caveat that the complexity of this syndrome cannot be captured in a single pre-clinical model, our findings support the biological plausibility of the epidemiological connections between vulvar pain and allergies and suggest that otherwise unexplained vulvar pain accompanied by tissue mast cell abnormalities may potentially be alleviated by local depletion of mast cells.

## 4. Materials and Methods

### 4.1. Animal Usage

Eight to twelve-week-old female ND4 Swiss mice (Harlan Laboratories, Indianapolis, IN, USA) were housed in Macalester College’s animal facility, which maintains 12-h light/dark cycle room conditions. Mice are able to access food and water at all times. Macalester College’s animal ethics committee, the Institutional Animal Care and Use Committee, approved all experimental protocols (B16Su1 approved 1 July 2016).

### 4.2. DNFB Administration: Sensitization and Challenge

#### 4.2.1. Labiar and Vaginal Canal Challenges

ND4 female Swiss mice were shaved on the right flank (15 mm × 15 mm) 1 or 4 days prior to sensitization depending on whether or not sensitivity measurements were taken during that experiment. A 5 mm × 5 mm area of labiar skin was shaved for labiar allergen application. All mice were sensitized with a topical application of 25 µL of 0.5% DNFB (Sigma-Aldrich St. Louis, MO, USA, #D1529-10ML) dissolved in 0.9% saline on the shaved flank. Four days after sensitization, mice were challenged for 10 consecutive days on the shaved labiar skin (10 µL per labium) or in the vaginal canal with a total of 20 µL of 0.3% DNFB dissolved in 0.9% saline. For vaginal canal challenges, mice were gently restrained in a belly-up position, while placing a P20 pipette with a 0.1–20 µL pipette tip by the vaginal opening and gently pipetting the solution into the vaginal canal. To minimize leakage and ensure that the full volume has been administered into the vaginal canal, mice were held in position for an additional 10 s.

#### 4.2.2. THC Treatment

One day after the 10th DNFB vaginal canal challenge, previously sensitized mice were treated with 50 µg of THC (Sigma-Aldrich, St. Louis, MO, USA; methanol evaporated) dissolved in 20 µL of 0.9% saline in the vaginal canal using the same vaginal canal challenge procedure detailed above. Olive oil (Sigma-Aldrich, St. Louis, MO, USA) was added to the THC solution one drop at a time to help THC dissolve into 0.9% saline. THC was administered daily for 6 consecutive days.

#### 4.2.3. Electronic Von Frey Measurements

Mechanical tactile sensitivity was measured in the labiar region, specifically the ano-genital ridge, using an Electronic von Frey Anesthesiometer (IITC Corporation, Woodland Hills, CA, USA) and techniques established and refined by our research group [48]. Two baseline withdrawal thresholds were taken 1 and 2 days before hapten exposure. Mice were assigned to treatment groups to ensure similar average baseline values between treatment groups (±0.01). Sensitivity was assessed at serial timepoints after the last DNFB challenge or last THC treatment by one investigator who remained blinded to treatment. To calculate percent decrease from baseline, average post-challenge withdrawal values were subtracted from average baseline values for each mouse. Percent decrease from baseline values were averaged across each treatment group. We defined the threshold for pain sensitivity as a 33% decrease in withdrawal force from baseline to treatment [8,9,49].

#### 4.2.4. Tissue Collection

Serum, iliac lymph nodes, spleen, flank tissue, labiar tissue, and vaginal canal were collected from mice euthanized by 100% CO_2_ inhalation at various predetermined experimental timepoints. Fat was scraped off the flank and labiar tissue samples and half of the vaginal canal furthest from the introitus was discarded. Tissue samples were flash-frozen in liquid nitrogen and stored at −80 °C.

#### 4.2.5. RNA Extraction and Quantitative Real-Time Reverse Transcriptase PCR (qRT-PCR)

Total RNA from vaginal canal tissue and labiar skin with fat removed (Total RNA Mini Kit, Midwest Scientific, St. Louis, MO, USA), was eluted with RT-PCR grade water, and quantified in ng/µL RNA with a Nanodrop ND-1000 Spectrophotometer (ThermoScientific, Wilmington, DE, USA). The RNA was then used in a semi-quantitative reverse-transcriptase polymerase chain reaction (sqRT-PCR) assay to calculate relative transcript abundances of *Interleukin-6* (*IL-6*; Mm00446190_m1) and *interferon-γ* (*IFN-γ*; Mm01168134_m1)—both of which were normalized to the expression of our housekeeping gene, beta-2-microglobulin (*B2M*; Mm0043772_m1). 100 ng of RNA per reaction was reverse-transcribed in a 2720 Thermal Cycler (Life Technologies, Carlsbad, CA, USA) using the Superscript III First-Strand Synthesis System (Life Technologies). Relative transcript abundance was determined via qRT-PCR using TaqMan Gene Expression Assay Primer/Probe Sets and TaqMan MasterMix (Life Technologies in a calibrated StepOnePlus^TM^ Real Time PCR System). Gene expression was analyzed using the 2^−∆∆*C*t^ method by normalizing to *β-2m* and calculating as fold expression over controls [50].

#### 4.2.6. Immunofluorescent Staining of Mast Cells and Nerves

Flash-frozen labiar tissue samples were embedded in Tissue-Tek Optimal Cutting Temperature compound (Sakura Finetek, Torrance, CA, USA; #4583) and cryosectioned into 12 µm sections on a Leica CM1860 (Wetzlar, Germany). Sections were then rehydrated in phosphate buffered saline (PBS) and fixed in 4% paraformaldehyde (Sigma-Aldrich, St. Louis, MO, USA; #158127; pH 8.5). For mast cell staining, tissues were washed in PBS, permeabilized for 30 min with 0.1% Triton X-100 (Sigma-Aldrich, St. Louis, MO, USA; #X100) in PBS, and blocked for one hour in 5% normal donkey serum (Jackson ImmunoResearch Laboratories, West Grove, PA, USA; #017-000-121) in PBS. Slides were then incubated for one hour with Fluorescein Avidin D (Vector Laboratories, Burlingame, CA, USA; # A-2001; 1:1000) in blocking solution. For nerve staining, fixed tissue sections were stained with a primary rabbit polyclonal antibody against calcitonin gene related peptide (CGRP; Abbiotec, San Diego, CA, USA; 1:500) and AlexaFluor 488-conjugated secondary antibody (Thermo Fisher Scientific, Wilmington, DE, USA; 1:1000) as previously described [8,9]. Stained slides were cover-slipped with VECTASHIELD Antifade Mounting Medium with DAPI (Vector Laboratories; Burlingame, CA, USA; H-1200) and left at room temperature for 30 min before freezing at −20 °C.

#### 4.2.7. Confocal Imaging

Sections were imaged using a laser scanning confocal microscope. Composite images of ten optical 1 µm sections projected on the *z*-axis were analyzed using FluoView FV1000 image analysis software (Olympus Corporation, Center Valley, PA, USA) or ZEN 2.1 Imaging software Version 2.1.57.1000 (Carl Zeiss AG, Oberkochen, Germany). Mast cell density and CGRP^+^ nerve density were determined by fluorescent pixel intensity measurements. Average intensities were calculated from tissue sections pooled from 3–6 mice from each treatment group from 1–2 independent experiments. For mast cells, four representative 5000 µm^2^ regions of interest were taken in each of three sections per slide per mouse and 3–6 mice per treatment group. The sum of the values from the four regions of interest were taken and divided by 5000 µm^2^ to give a value of average fluorescent intensity/µm^2^ for each section quantified. For CGRP^+^ nerves, four representative 5000 µm^2^ regions of interest as well as one blank region were measured for three sections per each slide. Readings from the blank region were subtracted from the average of the four representative sections and the resulting value divided by 5000 µm^2^ to give an average fluorescent intensity/µm^2^ for each section quantified. Each treatment value was then divided by the average control value, and plotted on a fold expansion graph using PRISM 6.0 (GraphPad, San Diego, CA, USA).

#### 4.2.8. Quantification of Local Eosinophil Peroxidase (EPO) and Myeloperoxidase (MPO) Activity

During tissue collection, labiar tissue was weighed, placed in a weight-based volume of 0.5% hexadecyl trimethylammonium bromide (HTAB) buffer (Sigma-Aldrich, St. Louis, MO, USA), and stored at −80 °C. After freezing the tissue for at least 24 h, samples were thawed and four times the original amount of HTAB buffer was added. The samples were homogenized, sonicated, freeze-thawed, re-sonicated, and centrifuged before they were incubated in a substrate solution (16 mmol/L o-phenylenediamine (OPD; Sigma-Aldrich, St. Louis, MO, USA), 50 mM Tris-HCl buffer, and 0.01% H_2_O_2_ for 30 min. Absorbance was measured at 490 nm. Eosinophil peroxidase (EPO) levels were normalized to tissue weight and optical density (OD) values per gram of wet tissue were calculated. For MPO measurements, labiar tissue, upon extraction, was placed in a solution of 50 mM K_2_HPO_4_ buffer (pH 6.0; Sigma-Aldrich, St. Louis, MO, USA) with 0.05% HTAB and stored at −80 °C. At least 24 h later, the tissue was thawed and homogenized in HTAB buffer. The 5×-diluted homogenate was then sonicated thrice, freeze-thawed thrice, re-sonicated, and centrifuged. After a 20-min incubation in 50 mM phosphate buffer (pH 6.0; Sigma-Aldrich, St. Louis, MO, USA) with 0.025% H_2_O_2_ and 0.167 mg/mL o-dianisidine dihydrochloride (Sigma-Aldrich, St. Louis, MO, USA) at room temperature in the dark, absorbance was measured at 450 nm. Myeloperoxidase (MPO) levels were normalized to tissue weight and OD/g of wet tissue calculated.

#### 4.2.9. Lymph Node Processing and Flow Cytometry

Harvested iliac lymph nodes were passed through a 70 μm strainer, washed with staining buffer (2% fetal bovine serum in 1× PBS; Serum Source International, Charlotte, NC, USA) and stained with antibodies for immunophenotyping (1:100 dilution; Table 1) in the presence of Fc block (2.4G2; BioXCell) for 30 min on ice, and analyzed on a BD LSR Fortessa X-20 (Becton Dickinson, Franklin Lakes, NJ, USA). Acquired data were analyzed using FlowJo software (Version 10, FlowJo LLC, Ashland, OR, USA). Dead cells were excluded using Ghost Dye-BV510 (Tonbo Biosciences, San Diego, CA, USA).

#### 4.2.10. Flank Skin Processing and Flow Cytometry

We used DNFB-challenged flank tissue rather than labiar skin or vaginal canal samples in order to obtain enough cells for analysis. To prepare flank tissue samples, ND4 female Swiss mice were shaved on the back (15 mm × 15 mm) and both flanks (15 mm × 15 mm) 1 day prior to sensitization. All mice were sensitized with a topical application of 25 µL of 0.5% DNFB (Sigma-Aldrich, St. Louis, MO, USA; #D1529-10ML) dissolved in 0.9% saline on the shaved back. Four days after sensitization, mice were challenged on both shaved flanks with 25 µL (50 µL total) of 0.3% DNFB dissolved in 0.9% saline or 0.9% saline alone daily for 10 consecutive days. Leukocytes were isolated from flank skin samples from CO_2_-euthanized mice as previously described [51] 1 day after the cessation of challenges. Cells were washed, blocked for five minutes with 50 μL of supernatant from a 2.4G2 hybridoma cell line (BioXCell, West Lebanon, NH, USA), stained with fluorochrome-conjugated monoclonal antibodies (Table 2) at 1:100 dilutions for 30 min at 4 °C. and analyzed using an LSR Fortessa X-20 (Becton Dickinson, Franklin Lakes, NJ, USA) flow cytometer. Acquired data were analyzed using FlowJo software (Version 10, FlowJo LLC, Ashland, OR, USA). Dead cells were excluded using Ghost Dye-BV510 (Tonbo Biosciences, San Diego, CA, USA).

#### 4.2.11. IgE Enzyme-Linked Immunosorbent Assay (ELISA)

Serum samples were collected at various timepoints. IgE content in the serum was measured using an IgE ELISA kit (Bethyl Laboratories, Montgomery, TX, USA) after serum was diluted 1:20 in dilution buffer provided with the kit. Absorbances were recorded with a PowerWave XZ microplate spectrophotometer (Biotek Instruments, Winooski, VT, USA) and IgE concentration in each sample was calculated according to the manufacturer’s instructions.

#### 4.2.12. Data Analysis and Visualization

Data were analyzed in Excel 16.16 (Microsoft, Redmond, WA, USA), graphed using PRISM 6.0 (GraphPad, San Diego, CA, USA) and statistically analyzed in JMP 11.2 software (SAS, Cary, NC, USA). One-way ANOVA and post hoc Tukey HSD analyses were used to determine statistical differences between treatment groups and timepoints. Grubbs’ test was used to exclude any outliers. Statistical significance was defined as *p* < 0.05.

## Figures and Tables

**Figure 1 ijms-20-02163-f001:**
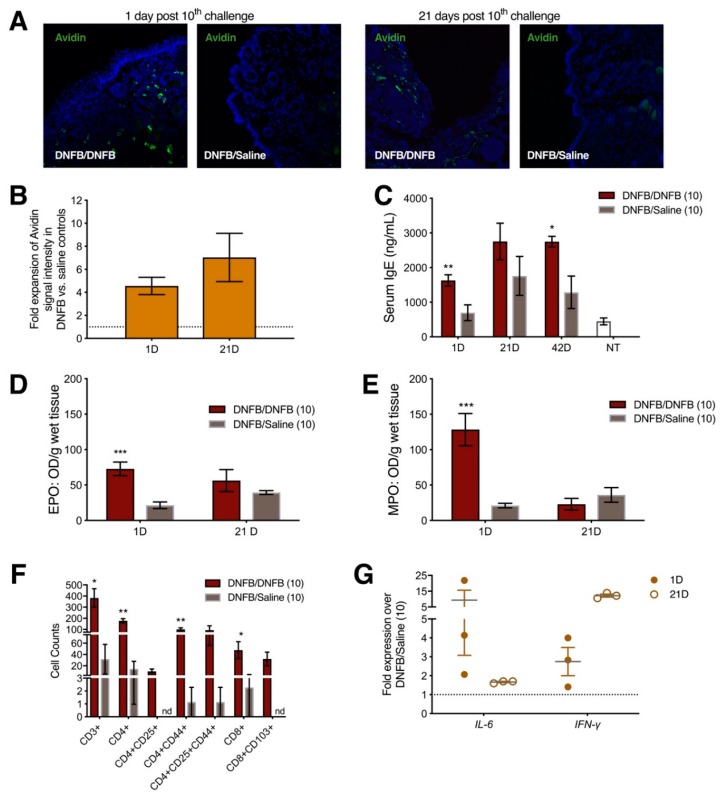
Repeated topical dinitrofluorobenzene (DNFB) exposure on the labiar skin in previously sensitized ND4 mice results in increased abundance of local mast cells, higher levels of *IL-6* and *IFN-γ* transcripts, elevated serum IgE, greater myeloperoxidase and eosinophil peroxidase activity, and increased numbers of infiltrating T cells. (**A**) Representative sections at 200x magnification showing FITC (fluorescein-isothiocyanate)-Avidin-stained mast cell granules in labiar tissues of DNFB vs. saline-challenged mice 1 day and 21 days after 10 challenges; (**B**) Relative intensity of Avidin-stained mast cell granules in the labiar skin of DNFB vs. saline-challenged mice 1 day and 21 days after 10 labiar DNFB challenges. *n*= 3; (**C**) Total IgE levels in serum measured 1 day, 21 days, and 42 days after 10 challenges with DNFB or saline in previously sensitized mice and in age-matched untreated mice (NT). ** and * represent *p*-values of *p* < 0.01 and *p* < 0.05 respectively. *n* = 3–6; (**D**) Eosinophil peroxidase and (**E**) myeloperoxidase levels in the labiar tissue, normalized to tissue weight, 1 and 21 days after 10 DNFB or saline challenges. *** represents *p*-values of *p* < 0.001. *n* = 3–6; (**F**) CsD3^+^, CD4^+^, CD4^+^CD25^+^, CD4^+^CD44^+^, CD4^+^CD25^+^CD44^+^, CD8^+^, and CD8^+^CD103^+^ cell numbers in flank tissues derived from previously sensitized mice challenged with either DNFB or saline vehicle on the flank. ** and * represent *p*-values of *p* < 0.01 and *p* < 0.05 respectively. *n* = 6; nd = not detected; (**G**) Relative abundances of *IL-6* and *IFN-γ* mRNAs in the labiar tissue of DNFB- vs. saline-challenged, previously sensitized mice 1 day and 21 days after 10 DNFB challenges, each normalized to *β2-microglobulin* (housekeeping) mRNA levels. *n* = 3.

**Figure 2 ijms-20-02163-f002:**
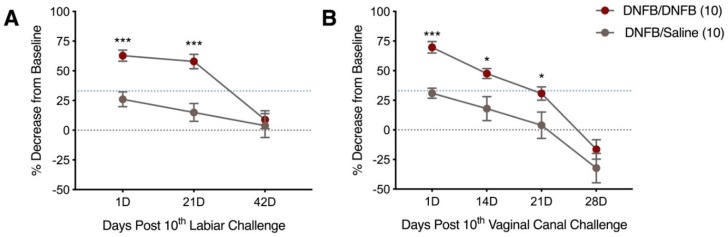
Ten daily DNFB challenges on the labiar skin and in the vaginal canal results in increased sensitivity that persists 21 days post tenth challenge. (**A**) Percent decrease from baseline measurements 1, 21, and 42 days after 10 labiar challenges, *n* = 18–27 and (**B**) 1, 14, 21, and 28 days post vaginal canal challenge. Only post-treatment withdrawal responses that show a 33%, represented by blue dotted line, or greater decrease from baseline withdrawal thresholds are considered hyperalgesic. *** and * represent a *p*-value of *p* < 0.001 and *p* < 0.05 respectively *n* = 9–18.

**Figure 3 ijms-20-02163-f003:**
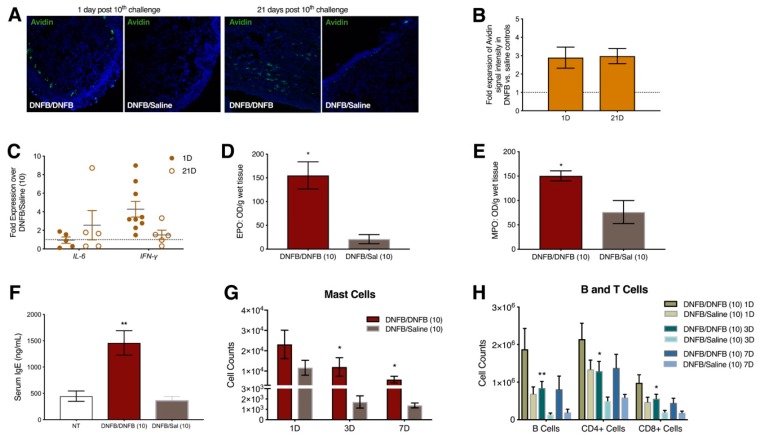
Repeated DNFB exposure in the vaginal canal provokes increased local mast cell density, upregulation of *IL-6* and *IFN-γ* expression, increased IgE concentration in the serum, and increased numbers of B cells, T cells, and mast cells in the iliac lymph nodes. (**A**) Representative immunofluorescent images at 200X magnification of FITC-Avidin stained mast cells in vaginal canal tissue of DNFB and saline challenged mice 1 day and 21 days post 10th challenge; (**B**) Relative fluorescent intensity of DNFB challenged mice 1 day and 21 days post 10 challenges compared to saline challenged mice. *n* = 3; (**C**) Relative transcript abundance of *IL-6* and *IFN-γ* mRNAs in the vaginal canal tissue normalized to *β2-microglobulin* (housekeeping) mRNA levels. *n* = 5–9; Local (**D**) eosinophil peroxidase and (**E**) myeloperoxidase levels in the vaginal canal, normalized to tissue weight, 1 day after 10 DNFB or saline challenges. * represents a *p*-value of *p* < 0.05. *n* = 3; (**F**) Total IgE levels in the serum measured 1 day after 10 challenges with DNFB or saline in previously sensitized mice and in age-matched untreated mice (NT). ** represents a *p*-value of *p* < 0.01. *n* = 4–6; Cell counts of (**G**) mast cells, and (**H**) B cells, CD4^+^ T cells, and CD8^+^ T cells in the iliac lymph nodes 1 day, 3 days, and 7 days post 10th challenge. ** and * represent a *p*-value of *p* < 0.01 and *p* < 0.05 respectively. *n* = 6.

**Figure 4 ijms-20-02163-f004:**
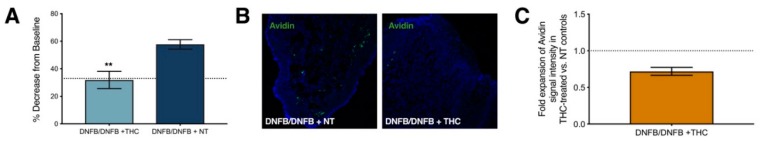
Repeated local Δ^9^-tetrahydrocannabinol (THC) application after ten DNFB challenges in the vaginal canal results in decreased sensitivity and local mast cell densities compared to controls. (**A**) Sensitivity measured 1 day after 10 DNFB challenges and either 6 THC treatments or no treatment (NT). Only post-treatment withdrawal responses that show a 33%, represented by the dotted line, or greater decrease from baseline withdrawal thresholds are considered hyperalgesic. ** represents a *p*-value of *p* < 0.01. *n* = 17–18; (**B**) Representative immunofluorescence images at 200x magnification showing fluorescent FITC-Avidin-stained mast cell granules in vaginal canals of THC treated and not treated mice; (**C**) Relative intensity of Avidin-stained mast cell granules in vaginal canals of THC treated mice calculated as fold change over non-THC treated mice. *n* = 3.

**Figure 5 ijms-20-02163-f005:**
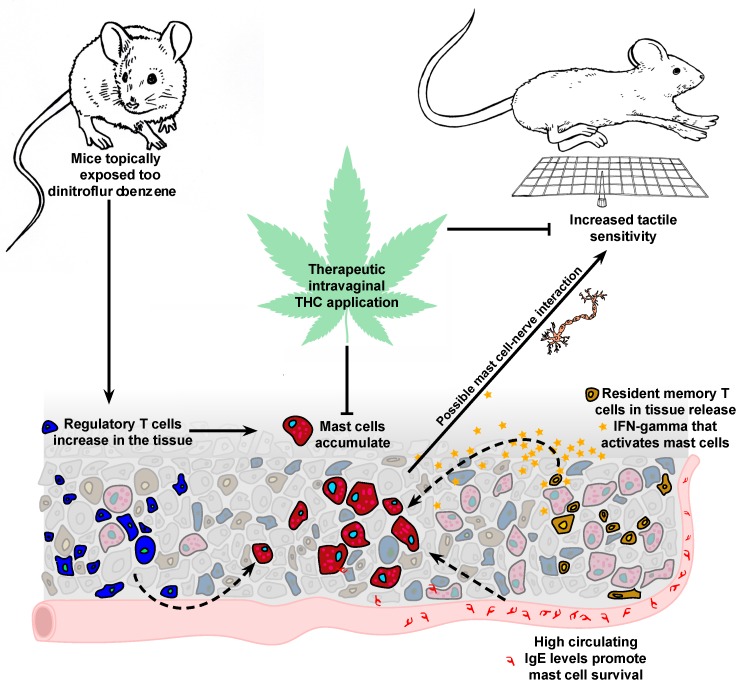
Repeated DNFB exposures on the skin lead to accumulation of regulatory T cells, resident memory T cells and mast cells. Mast cell increases are correlated with increased tactile sensitivity; therapeutic intra-vaginal THC application reduces mast cell density and tactile pain. Solid arrows indicate the hypothesized mechanisms proposed in this model. Dotted lines indicate previously published experimental observations of the contributions of regulatory T cells, IFN-γ and circulating IgE to mast cell recruitment, survival and activity in the tissue. T-bars indicate suppression of mast cell accumulation and tactile sensitivity following THC application. Artwork by Amy Pelz.

**Table 1 ijms-20-02163-t001:** Fluorochrome conjugated monoclonal antibody panel for flow cytometry of hapten treated draining LN samples used to identify distinct cell lineages.

Antigen	Color	Clone	Host	Stable Public Identifier	Vendor
CD3	BV421	145-2C11	Armenian Hamster	AB_2562556	BioLegend
CD4	PECy7	RM4-5	Rat	AB_2621829	Tonbo Biosciences
CD8	APC	53-6.7	Rat	AB_2621550	Tonbo Biosciences
CD19	BV785	6D5	Rat	AB_11218994	BioLegend
Ckit	PE	ACK2	Rat	AB_2621506	Tonbo Biosciences
FcεR1	FITC	MAR-1	Hamster	AB_10829796	Miltenyi Biotec
Ghost Dye	BV510	-	-	-	Tonbo Biosciences

**Table 2 ijms-20-02163-t002:** Fluorochrome conjugated monoclonal antibody panel for flow cytometry of hapten treated flank skin samples used to identify distinct cell lineages.

Antigen	Color	Clone	Host	Stable Public Identifier	Vendor
CD45.1	APC efluor 780	A20	Mouse	AB_1582228	Thermo Fisher
CD45.2	APC efluor 780	104	Mouse	AB_1272175	Thermo Fisher
CD3	BUV395	145-2C11	Armenian Hamster	-	BD Biosciences
CD19	BV786	6D5	Rat	AB_11218994	BioLegend
CD25	BV650	PC61	Rat	AB_11125760	BioLegend
CD8	APC	53-6.7	Rat	AB_2621550	Tonbo Biosciences
CD4	PerCP Cy5.5	RM4-5	Rat	AB_2621876	Tonbo Biosciences
CD44	efluor 450	IM7	Rat	AB_1272250	Thermo Fisher
CD103	FITC	2E7	Armenian Hamster	AB_10709438	BioLegend
Ghost Dye	BV510	-	-	-	Tonbo Biosciences

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
