# Peer review of "Tetrahydrocannabinol Reduces Hapten-Driven Mast Cell Accumulation and Persistent Tactile Sensitivity in Mouse Model of Allergen-Provoked Localized Vulvodynia"

_ijms, 2019, doi:10.3390/ijms20092163_

Round 1

Reviewer 1 Report

This work represents an important step in the study of chronic pain/neuropathy, particularly in relation to vulvodynia. Previous work by this same group established the presence of increased nerve growth and infiltration of mast cells into the affected tissue area following oxazolone exposure and therapeutic responsiveness using secretagogue compound 48/80. In this current study, they have utilized a method of priming, using the flank, prior to exposure of the labia. They have confirmed that localized sensitivity/mast cell infiltration is not solely the result of the vehicle used previously, and, finally, that topical treatment of THC to the local area reduces mast cell density and decreases sensitivity to pain, indicating that THC could be a useful treatment for vulvodynia and perhaps other chronic pain/neuropathic disorders.

I am happy with the work as it is presented here.

Author Response

Thank you for your time and feedback.

Reviewer 2 Report

The authors demonstrated the accumulation of mast cells in the labial tissue in the dinitrofluorobenzene (DNFB)-induced vulvodynia model. Treatment with tetrahydrocannabinol resulted in attenuated persistent tactile sensitivity, which was accompanied by decrease in the number of the local mast cells. The experiments were well designed and performed. Their findings at least in part support their hypothesis that local mast cells should play critical roles in the persistent tactile sensitivity induced by repeated treatment with DNFB. However, because a series of studies including this manuscript might lack the direct evidence, such as loss of vulvodynia in the mice lacking tissue mast cells, and mast cell stabilizers, such as cromolyn, were found to have no therapeutic effects, the authors should not emphasize too much the roles of the local mast cells in vulvodynia. It is possible that increased tactile sensitivity should be induced in mast cell-independent manner.

The basal levels of serum IgE concentrations in untreated mice should be described to evaluate the increase in the concentrations of IgE in this model.

Although the authors stated in the section Discussion that they have reported the early accumulation of regulatory CD4+CD25+ T cells in DNFB/saline-challenged skin that may serve to recruit mast cells to the site of the reaction, they did not cite any references. In Figure 5, regulatory T cells recruit mast cells although no related results could be found in this manuscript

Author Response

Point 1: Their findings at least in part support their hypothesis that local mast cells should play critical roles in the persistent tactile sensitivity induced by repeated treatment with DNFB. However, because a series of studies including this manuscript might lack the direct evidence, such as loss of vulvodynia in the mice lacking tissue mast cells, and mast cell stabilizers, such as cromolyn, were found to have no therapeutic effects, the authors should not emphasize too much the roles of the local mast cells in vulvodynia. It is possible that increased tactile sensitivity should be induced in mast cell-independent manner.

Response 1: Reviewer 2 asked us to be cautious about putting forward mast cells as important regulators of vulvodynia since the mast cells stabilizer sodium cromolyn is not an effective therapy for it.  We have clarified in Lines 267-272 that mast cell accumulation rather than degranulation may be correlated with the effects we see here i.e. the contributions of mast cells to tactile-sensitivity may be degranulation independent and therefore refractory to cromolyn which prevents degranulation and stabilizes mast cells.  We are careful to use the word “potential” in Lines 296-302 when we talk about the possibility of mast cell-targeted therapies that may be used broadly beyond vulvodynia.

Point 2: The basal levels of serum IgE concentrations in untreated mice should be described to evaluate the increase in the concentrations of IgE in this model.

Response 2: We have edited the graphs in Figures 1C and 3F to include the baseline levels of serum IgE in untreated, age-matched female ND4 mice as requested by reviewer 2.  We have reordered and rephrased lines 69-73 to further clarify the “Results” text that refers to Figure 1C.

Point 3: Although the authors stated in the section Discussion that they have reported the early accumulation of regulatory CD4+CD25+ T cells in DNFB/saline-challenged skin that may serve to recruit mast cells to the site of the reaction, they did not cite any references. In Figure 5, regulatory T cells recruit mast cells although no related results could be found in this manuscript

Response 3: Reviewer 2 commented that we had not cited a reference to support our statement that regulatory T cell accumulation in the airways is a pre-requisite for mast cell recruitment therein.  We respectfully submit that we had, in fact, cited a reference (#33; Jones et al, 2010) that reported this finding.  We have, in this revision, rephrased the text in Lines 218-219 to further clarify this. Jones et al. (33) reported that regulatory T cells were necessary for mast cell infiltration following allergen exposure and here we find both regulatory T cells and mast cells are present after 10 DNFB exposures.  We therefore hypothesize in our model figure (Figure 5) that the relationship observed by Jones et al. underlies the observations we make here.  We have not previously reported DNFB-driven regulatory T cell accumulation in mouse tissues.